# Soil Quality Assessment in Farmland of a Rapidly Industrializing Area in the Yangtze Delta, China

**DOI:** 10.3390/ijerph191912912

**Published:** 2022-10-09

**Authors:** Xiangling Zhang, Yan Li, Genmei Wang, Huanchao Zhang, Ruisi Yu, Ning Li, Jiexiang Zheng, Ye Yu

**Affiliations:** 1Co-Innovation Center for the Sustainable Forestry in Southern China, Nanjing Forestry University, Nanjing 210037, China; 2College of Forestry, Nanjing Forestry University, Nanjing 210037, China

**Keywords:** farmland, soil fertility, heavy metals, comprehensive assessment

## Abstract

The comprehensive quality assessment of farmland soil is critical for agricultural production and soil ecological protection. Currently, there is no systematic method for conducting a comprehensive quality assessment of farmland soil; subsequently, as the most developed economic area in China, the comprehensive quality assessment of farmland soil in the Yangtze River Delta is lacking. We chose the farmland soil of Suzhou city as the research object. The soil fertility index (SFI) and soil environment index (SEI) were calculated with the membership function and Nemerow index. Finally, the comprehensive assessment of soil quality was achieved with the TOPSIS model. The results showed that the average values of soil pH, SOM, TN, AHN, AP, and AK were 6.44 (slightly acidic), 28.17 g/kg (medium), 1.63 g/kg (rich), 118.16 mg/kg (medium), 38.31 mg/kg (rich), and 160.63 mg/kg (rich), respectively. For the concentrations of heavy metals, including Cr, Ni, Cu, Zn, Cd, and Pb, in 122 soil samples, the percentages exceeding the background values of Jiangsu province were 5.74%, 8.20%, 8.20%, 10.66%, 86.07%, and 84.43%, respectively. Cd and Pb were the main heavy metal pollutants on farmlands. The soil samples with SFI values below the medium level (SFI < 0.6) accounted for 44.26%, and samples with SEI values below the medium level (SEI < 0.6) accounted for 13.12%. The values of the soil quality index (SQI) ranged from 0.171 to 0.996, with an average SQI value of 0.586 (very poor—V), and approximately half of the farmland soil quality in Suzhou city needed to be further improved. In a word, this study provides a theoretical basis and scientific support for the quality assessment and rational utilization of farmland soil.

## 1. Introduction

Soil is the most important part of the ecosystem, and provides the material basis for plant growth and development [1,2]. More importantly, soil represents the foundation for agricultural activities, which are directly related to the improvement of people’s livelihood and social stability [3,4]. In China, with the economic growth, industrialization, urbanization, agricultural intensification, and global climate change, the contradictions between the population, resources, environment, and development have been worsening [5,6], and the total area of farmland has also decreased sharply. On the other hand, farmland soil is facing many quality problems, such as soil erosion, soil fertility decline, soil acidification, salinization, and soil pollution, which severely restrict the supply capacity of agricultural products [7,8]. Therefore, monitoring and assessing soil quality is of great importance for agricultural production and soil ecological protection.

Soil quality can be defined as “the ability of soil to sustain plant and animal productivity, maintain or enhance water and air quality, and to support human health and habitation” [9,10]. Thus far, the comprehensive assessment system for soil quality is still lacking [11]. In the assessment of soil quality, the method combining fertility indicators and environmental indicators can not only reflect the ability of soil to provide nutrients to plants and generate biomass, but also reflect the pollution status of the soil environment. For a long time, soil fertility quality and soil environmental quality have been independent systems in the soil quality assessment. In recent years, a few scholars began to explore the comprehensive assessment methods of soil quality. Li et al. [12] established a multi-criteria assessment system for arable land resources by using the soil comprehensive fertility index and Nemerow index. Fan et al. [13] calculated the soil environmental index (SEI) and soil fertility index (SFI) by improving the Nemerow index, and, finally, obtained the soil quality index (SQI) by using the multi-criteria quantitative method. Yang et al. [14] combined soil fertility with heavy metal pollution by using the membership function and TOPSIS model, and realized the comprehensive assessment of soil quality under different land uses. However, these explorations had certain limitations. For example, Li et al. [12] and Yang et al. [14] reflected the soil environmental level through the use of the Nemerow index; however, its SQI classification standard is only applicable to certain types of studies, and is not suitable for comparison with other studies. They all failed to form a universally applicable and objective assessment system and their assessment systems would be limited by the assessment object and assessment scope during the utilization process. To address these research limitations, we attempt to achieve the standardization of environmental indicators without the Nemerow index, but through the membership function and geo-accumulation index, which could make the data more referential and comparative. Moreover, the soil comprehensive quality index obtained by using the entropy-based TOPSIS model would be more objective.

Like many cities in developing countries and emerging economies, farmland and soil resources in the Yangtze River Delta are facing great pressure with the rise of industrialization and rapid urbanization. We chose Suzhou city of the Yangtze River Delta as the research area. The Yangtze River Delta is one of the most densely populated and urbanized regions in China. Moreover, Suzhou is also a typical city transitioning from traditional agriculture to modern agriculture, modern industry, and advanced manufacturing industry in the region [15,16]. Additionally, its agro-ecosystem is facing not only heavy metal pollution, but also soil quality problems, such as nutrient content decline [17,18]. Meanwhile, the comprehensive assessment of farmland soil quality in the Yangtze River Delta is also lacking.

The objectives of this study are: (1) to establish the membership function for soil environmental indicators by using the geo-accumulation index; (2) to calculate the SFI and SEI by combining the Nemerow index and the membership function; (3) to calculate the SQI by combining the SFI and SEI in the TOPSIS model; (4) to determine recommendations from the perspective of agricultural farming. The results of our study could provide scientific guidance for farming management, pollution prevention and control, and the green and sustainable development of agriculture.

## 2. Materials and Methods

### 2.1. Study Area

Suzhou is located in the southeast of Jiangsu province and the central Yangtze River Delta, China. The city of Suzhou consists of six districts (Gusu district, Huqiu district, Wuzhong district, Xiangcheng district, Wujiang district, and the Suzhou industrial park) and four county-level cities (Zhangjiagang, Changshu, Taicang, and Kunshan). The total area of the city is 8657.32 km^2^, with more than 20,000 rivers of different sizes and more than 300 large and small lakes. The areas of rivers, lakes, and mudflats account for 36.6%. In addition to its superior geographical location, Suzhou also has fertile land and superior natural conditions, and its climate belongs to the sub-tropical monsoon marine climate, which is mild and humid with four distinct seasons [19]. The natural vegetation in the Yangtze River Delta is dominated by deciduous and evergreen forests [20]. The delta has many soil types, with the major ones being paddy soil, red soil, fluvo-aquic soil, and coastal saline soil [21]. Paddy soil is the most widely distributed soil in the delta, and the main soil type of Suzhou is hydro-morphic paddy soil [22]. Rice and wheat are the main crops in Suzhou, especially rice, which is the dominant crop in Suzhou and the crop with the largest sown area. In 2020, the sown areas of grain crops in Suzhou reached 1195.80 km^2^, and the total grain output reached 888,100 tons. The annual vegetable planting areas remained stable at 667 km^2^ [19]. More importantly, Suzhou is a large industrial city with 35 industrial categories involving 167 middle industrial categories and 489 industrial sub-categories. It is one of the cities with the most complete industrial systems in China. The electronic information industry, equipment manufacturing, chemical and petroleum industry, metallurgical industry, light industry, and textile industry are the six leading industries in Suzhou [16].

### 2.2. Sample Collection and Laboratory Analysis

Soil sampling was performed in May 2021. A total of 122 surface soil samples (0–20 cm) was obtained from the farmland in Suzhou (including paddy, irrigated, and dry fields) (Figure 1). Considering the impact of urbanization on the soil quality of farmland, farmland samples from nearby towns were added. At each site, we collected 5 sub-samples with 5 m as the diagonal and then mixed sub-samples together to form one soil sample. During sampling, we avoided field edges and removed gravel and plant debris. In addition, the geographic co-ordinates of each sample site were located with GPS. Collected soil samples were stored in clean plastic bags and transported back to the laboratory. Subsequently, the soil samples were air-dried at room temperature (25 °C in this case) to a constant weight, and impurities, such as stones and roots, were removed. All samples were sieved through nylon sieves (2 mm, 0.25 mm, 0.149 mm) and then stored for analysis.

The analysis of fertility indicator referenced the *soil agrochemical analysis* [23]. The soil samples were digested with four acid digestion methods (HCl-HNO_3_-HF-HClO_4_) [24]. In reference to the assessment of heavy metal pollution in the soil of Suzhou [25] and the selection of assessment indicators in previous studies [26,27], the heavy metals Cr, Ni, Cu, Zn, Cd, and Pb were selected as assessment indicators in this study. The contents of heavy metals were determined by using an inductively coupled plasma mass spectrometer (ICP-MS, ICAP Q, Thermo Fisher Scientific, Bremen, Germany). All reagents for the analysis were of superior grade, and the water used was ultra-pure. All vessels were previously cleaned by soaking in 10% nitric acid for more than 24 h and, finally, rinsed thoroughly with ultra-pure water before use. The method blanks, parallel samples, and national standard material (GBW07405) were included for quality assurance and quality control, and 20% of parallel samples was analyzed for each batch [28]. The detection limits (ICP-MS) of Cr, Ni, Cu, Zn, Cd, and Pb were 0.02–0.001 μg/L. The recovery rates of heavy metals were 87.1–106.0%.

### 2.3. Soil Quality Assessment Method

The corresponding membership function was selected to calculate the membership degree of each assessment indicator, and the soil fertility index (SFI) and soil environment index (SEI) were calculated with the modified Nemerow index. Eventually, the soil comprehensive quality index (SQI) was obtained by using the TOPSIS model according to the SFI and SEI of each sample site.

#### 2.3.1. Standardization of Soil Fertility Indicators

With reference to the selection of previous assessment indicators [29] and taking into account the cultivation tradition of Suzhou city, which mainly grows rice, the soil pH, soil organic matter (SOM), total N (TN), alkali-hydrolyzed N (AHN), available P (AP), and available K (AK) were selected as assessment indicators in this study. According to the standards in the second state soil survey of China [30] and the recommended grading standards for the soil fertility indicators of paddy soil in the Taihu Lake plain from the *indicators and assessment* of *soil quality* [31], the classification standards for soil fertility indicators in this study were determined (Table 1) to be grades I to V, representing: richer, rich, medium, lacking, and very lacking, respectively.

The relationships between soil fertility and crop growth effect, soil fertility, and crop yield were quantified with an appropriate membership function, the pH was suitable for the parabolic-type membership function, and SOM, TN, AHN, AP, and AK were suitable for the “S”-type membership function [32,33].

The parabolic-type membership function:(1)$$FI=\left\{\begin{array}{cc} 0.1 & x\leq L,x\geq U\\ 0.1+0.9\left(x-L\right)/\left(O_{1}-L\right) & L<x<O_{1}\\ 1.0 & O_{1}\leq x\leq O_{2}\\ 1.0-0.9\left(x-O_{2}\right)/\left(U-O_{2}\right) & O_{2}<x<U \end{array}\right.$$

The “S”-type membership function:(2)$$FI=\left\{\begin{array}{cc} 0.1 & x\leq L\\ 0.1+0.9\left(x-L\right)/U-L & L<x<U\\ 1.0 & x\geq U \end{array}\right.$$
where FI is the membership value of the soil fertility indicator; x is the measured value; U and L are the upper and lower limits of the function; $O_{1}$ and $O_{2}$ are the lower and upper limits of the optimal value of the function. The value range of the SFI was [0.1, 1] when the maximum value was 1, which meant that the indicator content was completely suitable for plant growth [34]. Combined with Table 1, the thresholds for the membership function of fertility indicators were determined (Table 2).

#### 2.3.2. Standardization of Soil Environmental Indicators

The geo-accumulation index $\left(I_{geo}\right),$ also known as, the Muller index, was use in the assessment of soil heavy metal pollution; this method could simultaneously reflect the geological factors and the effects caused by human activities [35]. $I_{geo}$ could be calculated as follows:(3)$$I_{geo}=\log_{2}\left(C_{n}/1.5B_{n}\right)$$
where $\;C_{n}$ is the measured value of heavy metal (mg/kg); $\;B_{n}$ is the geo-chemical background value of element n (mg/kg). The geo-chemical background value of Jiangsu province was used in this study [36].

Referring to the studies of Fazel Rahmanipour et al. [37] and Ya’nan Fan et al. [13], the relationship between $I_{geo}$ and soil environmental quality conformed to the inverse “S” membership function. The standardization of soil environmental indicators could be realized by using Equation (4).(4)$$EI=\left\{\begin{array}{cc} 0 & I_{geo}\geq I_{geo-max}\\ \left(I_{geo-max}-I_{geo}\right)/I_{geo-max} & 0<I_{geo}<I_{geo-max}\\ 1.0 & I_{geo}\leq 0 \end{array}\right.$$

Here, EI is the membership value of the soil environmental indicator, $I_{geo-max}$ is the value calculated with the ‘risk screening values’ of six heavy metals under different soil pH levels from the soil environmental quality risk control standard for the soil contamination of agricultural land in China [38] (Equation (4)). $I_{geo}\leq 0$ indicates that the soil heavy metal concentration was lower than the background value, un-polluted, and had the highest membership value (EI = 1); $0<I_{geo}<I_{geo-max}$ indicates that heavy metal had accumulated in the soil, but the heavy metal concentration was still lower than the ‘risk screening values’; $I_{geo}\geq I_{geo-max}$ indicates that the heavy metal concentration was greater than the ‘risk screening values’ and had the lowest membership value (EI = 0). After standardization with the membership function, the value range of EI was [0, 1]. The higher the EI value was, the lower the pollution degree of the heavy metal.

#### 2.3.3. Calculation of the SFI and SEI

The Nemerow index method proposed by Nemerow has been widely used in the field of environmental quality assessment. This method takes into account the extreme value and average value of the assessment object data set [39]. Scholars improved the Nemerow index method to assess soil fertility quality [40]; the results of their studies also indicated that the improved Nemerow index method could reflect the minimum factor law for crop growth in ecology, and it was more rigorous and sensitive than the traditional weighting method for the assessment of soil fertility [37,41].

The SFI and the SEI were calculated based on the modified NPI method [13] using Equations (5) and (6).
(5)$$SFI=\sqrt{\left(FI_{min}{}^{2}+FI_{ave}{}^{2}\right)/2}\;\;\;$$
(6)$$SEI=\sqrt{\left(EI_{min}{}^{2}+EI_{ave}{}^{2}\right)/2}$$

Here, $FI_{min}$ and $FI_{ave}$ are the minimum and average membership values of the soil fertility indicators at the sample site, respectively. $\;EI_{min}$ and $EI_{ave}$ are the minimum and average membership values of the soil environmental indicators, respectively. Thus, the effect of the worst soil fertility indicator and the worst soil environmental indicator on soil quality could be better reflected by replacing the maximum value in the traditional Nemerow index formula with $FI_{min}$ and $EI_{min}$.

#### 2.3.4. Soil Quality Comprehensive Assessment: An Entropy-Based TOPSIS Model

The technique for order preference by similarity to ideal solution (TOPSIS) is a comprehensive analysis method that can effectively solve decision-making problems with multi-standard and multi-attribute aspects [42]. Based on the dimensionless data matrix, the Euclidean distance between the assessment index and the optimal and worst solutions was calculated, and the closer the assessment object was to the optimal solution, the higher the score was, and, conversely, the lower it was. Due to the different units of each assessment indicator, the original data needed to be standardized in the calculation process of the TOPSIS. In this study, the values of the SFI and SEI were transformed into dimensionless values of [0, 1] with Equations (1)–(6).

In the comprehensive assessment, each assessment indicator had a different effect on the assessment result, so it was necessary to determine the weight of each indicator. The entropy weight method is an objective weighting method, using information entropy to determine the weight of indicators according to the dispersion degree of the original data of each indicator. Here, the SFI and SEI were weighted by using the entropy weight method.

Step 1: Obtain the entropy weight of each assessment indicator ($W_{i\;}$):(7)$$p_{ij}=\frac{S_{ij}}{\sum_{i=1}^{m}S_{ij}}$$
(8)$$e_{i}=-\frac{1}{\ln m}\sum_{i=1}^{m}P_{ij}\ln P_{ij}$$
(9)$$W_{i}=\frac{1-e_{i}}{\sum_{i=1}^{n}(1-e_{i})}$$
where m and n are the numbers of assessment samples and assessment indicators, respectively [43]. $S_{ij}$ is the standardized value of index  i in the object j, $p_{ij}$ is the weight of index i in the object j, and when $p_{ij}=0$, $P_{ij}\ln P_{ij}$ = 0, it is meaningless. $e_{ij}$ is the information entropy value.

Step 2: Calculate the SQI with the TOPSIS model [43,44]:(10)$$V_{ij}=W_{i}\times X_{ij}$$
(11)$$V_{i}^{+}=\left\{maxV_{ij}|i=1,2,\cdots ,m\right\}=\left\{V_{1}^{+},V_{2}^{+},\cdots ,V_{m}^{+}\right\}$$
(12)$$V_{i}^{-}=\left\{minV_{ij}|i=1,2,\cdots ,m\right\}=\left\{V_{1}^{-},V_{2}^{-},\cdots ,V_{m}^{-}\right\}$$
(13)$$D_{j}^{+}=\sqrt{\sum_{i=1}^{m}{\left(V_{ij}-V_{i}^{+}\right)}^{2}}$$
(14)$$D_{j}^{-}=\sqrt{\sum_{i=1}^{m}{\left(V_{ij}-V_{i}^{-}\right)}^{2}}$$
(15)$$SQI_{j}=\frac{D_{j}^{+}}{D_{j}^{+}+D_{j}^{-}}$$
where $X_{ij}$ is the standardized data matrix; $W_{i}$ is the weight of the index; $V_{i}^{+}$ is the optimal solution; $V_{i}^{-}$ is the worst solution; $D_{j}^{+}$ and $D_{j}^{-}$ are the distances from the assessment object j to the optimal solution and the worst solution, respectively. The smaller $D_{j}^{+}$ indicates that the closer the assessment object was to the optimal solution, the better the soil quality. The smaller $D_{j}^{-}$ indicates that the closer the assessment object was to the worst solution, the worse the soil quality. The SQI value also ranged from 0 to 1 (the higher, the better).

### 2.4. Spatial Interpolation Method

Referring to the studies of Chen et al. [45], Hou et al. [46], and Wu et al. [47], we used the inverse distance weighting (IDW) method to map the spatial distribution of soil assessment indicators and the spatial distribution of the SFI, SEI, and SQI.

The IDW method was developed based on Tobler’s first law of geography, which states that “everything is related to everything else, but near things are more related than distant things” [48]. The method calculates the prediction values of interpolated points through a weighted linear combination of known points. IDW is simple and intuitive, and interpolation could be calculated quickly. Currently, IDW has become one of the many spatial interpolation methods and a popular tool in geographic information systems (GISs). The interpolation function could be expressed as follows [46,49]:(16)$$Z\left(x\right)=\frac{\sum_{i=1}^{n}W_{i}Z_{i}}{\sum_{i=1}^{n}W_{i}}$$
(17)$$W_{i}=d_{i}^{-u}$$
where Z(x) is the predicted value of the interpolated point; $Z_{i}$ is the measured value of the known point; *n* is the number of the neighboring points; $W_{i}$ is the weight assigned to each sampling point, and, in this study, it was determined to be 2, which is most commonly used [50]; $d_{i}\;$ is the distance from each known point to the interpolated point; u is the value of the distance decay parameter, and the effect of known points on the weight of interpolated points decreased exponentially when increasing the distance.

## 3. Results

### 3.1. Characteristics of Soil Quality Indicators

The descriptive statistics of each assessment indicator in 122 soil samples are displayed in Table 3. The mean contents of soil pH, SOM, TN, AHN, AP, and AK were 6.44 (slightly acidic), 28.17 g/kg (medium), 1.63 g/kg (rich), 118.16 mg/kg (medium), 38.31 mg/kg (rich), and 160.63 mg/kg (rich), respectively. The coefficient of variation (CV) represents the dispersion degree of the assessment indicator in the spatial distribution [51]. According to Wilding [52], a low variation, moderate variation, and high variation are defined by values of CV ≤ 16%, 16 < CV ≤ 36%, and CV > 36%, respectively. The CVs for soil pH indicated a low variation, and for the SOM, TN, AHN, AP, and AK, they indicated a high variation. Furthermore, the CV of the soil AP was the largest at 165.95%. The high CVs for the contents of the SOM, TN, AHN, AP, and AK indicated that these nutrients in the topsoil differed greatly with respect to different areas, and were seriously affected by human activities.

The concentrations of the heavy metals Cr, Ni, Cu, Zn, Cd, and Pb in soil ranged from 17.79 to 493.09 mg/kg, from 7.89 to 244.97 mg/kg, from 6.19 to 49.03 mg/kg, from 23.20 to 167.99 mg/kg, from 0.076 to 0.644 mg/kg, and from 16.22 to 81.43 mg/kg, respectively. For the concentrations of the heavy metals Cr, Ni, Cu, Zn, Cd, and Pb in 122 soil samples, the percentages of exceeding the ‘risk screening values’ were 3.28%, 3.28%, 0%, 0%, 8.20%, and 0.82%, respectively; the percentages of exceeding the background values were 5.74%, 8.20%, 8.20%, 10.66%, 86.07%, and 84.43%, respectively. The average concentrations of the heavy metals Cr, Ni, Cu, and Zn were lower than the background values, while the mean values of Cd and Pb exceeded the background values by 1.56 and 1.31 times, respectively. This suggested that Cd and Pb were the main heavy metal pollutants in the farmlands of this study. Moreover, the concentrations of the heavy metals Cr, Ni, Cu, Zn, and Cd presented a high variation, except for Pb, which indicated a moderate variation.

### 3.2. Spatial Distribution of Soil Fertility Indicators

Figure 2 shows that the spatial distributions of all fertility indicators were un-even. Soil samples with pH < 4.5 (strongly acidic), 4.5–5.5 (acidic), 5.5–6.5 (weakly acidic), 6.5–7.5 (neutral), and 7.5–8.5 (weakly alkaline) accounted for 1.64%, 22.95%, 31.97%, 19.67%, and 23.77%, respectively. Obviously, the pH value of most soils in the north of Suzhou was higher than that of the soil in the south. The contents of the SOM, TN, and AHN showed similar spatial distribution characteristics (Figure 2 a–c). Referencing Table 1, the sample sites with a SOM content at lacking and very lacking levels accounted for 22.95% and 6.56%, respectively, and they were mainly distributed in Xiangcheng district, Taicang, and some areas of Kunshan. The percentage of soil samples with a TN content at or above the medium level was 81.14%. The sample sites with an AHN content at lacking and very lacking levels accounted for 16.39% and 18.03%, respectively, and they were mainly distributed in Xiangcheng district, Taicang, Kunshan, and some areas of Zhangjiagang. The sample sites with an AP content at the medium level and above reached 82.79%, and were mainly distributed in central Suzhou.

### 3.3. Spatial Distribution of Soil Heavy Metal Concentrations

Similarities in spatial distribution patterns were found between Cr and Ni and between Cu and Zn (Figure 3). High concentrations of Cr and Ni were mainly distributed in Xiangcheng district, Huqiu district, and the junction between Xiangcheng district and the Suzhou industrial park. For Cu and Zn, high-concentration samples were concentrated in Huqiu district, Gusu district, and Xiangcheng district. For Cd, high-concentration samples were concentrated in Huqiu district, Gusu district, and Zhangjiagang. For Pb, high concentrations were mainly concentrated in Huqiu district and its junction with Gusu district, Xiangcheng district, and Wuzhong district. Overall, the concentrations of six heavy metals in the farmland of Huqiu and Gusu districts were relatively high.

### 3.4. Pollution Status of Heavy Metals in Soils

The calculation of $\;I_{geo}$ showed (Figure 4) the average $I_{geo}$ of different heavy metals in the following order: Cd (0.07) > Pb (−0.25) > Ni (−1.09) > Zn (−1.14) > Cu (−1.39) > Cr (−1.56). According to the classification of the $I_{geo}$ values [53,54], the pollution levels of all six heavy metals were between un-polluted ($I_{geo}\leq 0$) and moderately to strongly polluted ($2<I_{geo}\leq 3$), and without higher-grade pollution. This suggested that Cd and Pb had a greater accumulation than other heavy metals in the soil. The soil samples with Cd fell into un-polluted to moderately polluted ($0<I_{geo}\leq 0$), where moderately polluted ($1<I_{geo}\leq 2$) accounted for 33.61% and 4.92%, respectively. Additionally, the soil samples with Pb fell into un-polluted to moderately polluted, where moderately polluted accounted for 13.11% and 1.64%, respectively. Cr, Ni, Cu, and Zn generally exhibited un-polluted levels in most soil samples.

### 3.5. Comprehensive Assessment Results

After the entropy weighting method, the weights of the SFI and SEI in this study were 0.686 and 0.314, respectively. The values of the SQI calculated with the TOPSIS model were the same as the values of the SEI and SFI, which were dimensionless values of 0 to 1. Referring to previous studies [13,42], the SFI, SEI, and SQI were classified into five grades in this study: excellent—I (
≥0.80); good—II (0.80–0.70); medium—III (0.70–0.60); poor—IV (0.60–0.40); and very poor—V (< 0.40). The distributions of the SFI, SEI, and SQI were drawn with the IDW method in ArcGIS 10.5 (Figure 5).

The SFI values ranged from 0.181 to 1.000, and the mean value was 0.612 (medium). The sample sites with SFI values at poor and very poor fertility levels accounted for 19.67% and 24.59%, respectively, and they were mainly distributed in central Suzhou and Kunshan, and Taicang cities. For the SEI, its value ranged from 0.333 to 1.000, and the average SEI value was 0.886 (excellent—I). Most soil samples (86.88%) of Suzhou city were at the medium and above environment levels, which suggested that most farmland soils were not polluted by heavy metals. It should be pointed out that the low values of the SEI were mainly concentrated in central Suzhou, especially in Huqiu district, where the soil environment was the worst.

The SQI values ranged from 0.171 to 0.996, with an average SQI value of 0.586 (very poor—V). The sample sites with SQI values at poor and very poor fertility levels accounted for 18.85% and 35.25%, respectively. Summary statistics (Figure 5c) also showed that the farmland of Suzhou city had a higher percentage of samples with concentrations below the medium fertility level. Overall, approximately half of the farmland soil quality in Suzhou city needed to be further improved, and the low values of the SQI were mainly concentrated in central Suzhou and most areas in Kunshan and Taicang. Meanwhile, the soil environmental condition of farmlands in Suzhou was better than the soil fertility condition. The spatial distribution of the SQI was similar to that of the SFI, and the weights determined with the entropy weighting method also showed that the high/low values of the SFI had a greater influence on the soil quality assessment results.

## 4. Discussion

### 4.1. Correlation Analysis between Different Soil Indicators

As shown in Figure 6, the SOM and TN (0.784), SOM and AHN (0.756), and TN and AHN (0.930) all showed strong positive correlations (*p* < 0.05), which may be related to the fact that the N nutrients in soil primarily exist in organic forms [55]. The soil pH was strongly negatively correlated (*p* < 0.05) with the TN (−0.59) and AHN (−0.66), which may be related to soil acidification caused by the excessive long-term application of nitrogen fertilizer [56]. Guo et al. [57] indicated that in agro-ecosystems, excessive un-used ammonia nitrogen produces NO_3−_ during nitrification, meanwhile increasing soil H^+^ and causing soil acidification. On the other hand, this negative correlation may be related to the high degree of industrialization in Suzhou, where nitrogen oxides discharged from industrial pollution cause acid deposition and lead to a decrease in soil pH [58].

For heavy metals, Cr and Ni (0.99), Cr and Cu (0.60), Ni and Cu (0.57), and Cu and Zn (0.88) all showed strong positive correlations (*p* < 0.05), and Cd and Pb (0.47) showed a significant positive correlation (*p* < 0.01). The correlation between different heavy metals may reflect that they originated from similar sources and had similar contamination levels [59,60].

The SOM, TN, AHN, and SFI all showed strong positive correlations. Additionally, the correlations between Cd and SEI (−0.75) and Pb and SEI (−0.58) were stronger. The SFI and SQI (0.961) were significantly strongly correlated, and the weight of the SEI was significantly higher than that of the SFI, which may suggest that the soil comprehensive quality was more closely related to soil fertility quality.

### 4.2. Soil Fertility Status and Heavy Metal Pollution

It is worth noting that both the single assessment index and the SFI values showed that there were still many farmlands with a low soil fertility, especially in Xiangcheng district, Kunshan, and Taicang. All soil fertility indicators, except for pH, showed high coefficients of variation, which may have been caused by the excessive and un-even application of fertilizers. For the AP, some samples with high concentrations could be found in central Suzhou. Although phosphorus is an essential element for plant growth and the lack of soil phosphorus can affect the yield of crops, too much soil phosphorus can lead to water eutrophication caused by soil phosphorus loss [61]. Studies have shown that the mutation point of soil phosphorus leaching loss was 39.9–90.2 mg/kg [62]. In our study, 7.32% of the samples had an AP content greater than 90.2 mg/kg, and the highest value even reached 404.62 mg/kg. From the perspective of protecting the ecological environment, the application of phosphorus fertilizer should be greatly reduced in some areas. The level of comprehensive soil fertility is closely related to the degree of co-ordination of various factors, and the co-ordination ability of each nutrient should be strengthened in field management to avoid too high or too low fertility.

The highest percentage of exceeding the ‘risk screening values’ was found for Cd, and $I_{geo}$ values of the heavy metals Cd and Pb showed higher accumulation in the topsoil, which indicated that pollution caused by Cd and Pb were relatively serious in the study area. Hou et al. [63] indicated that in the agro-ecosystem of the Yangtze River Delta region, the main source of the heavy metals Cd and Cu was irrigation water, and the main source of Zn and Pb was atmospheric deposition. Atmospheric deposition, fertilizers, and irrigation water contributed 67%, 32%, and 1% to Cd, and 84%, 3%, and 13% to Pb, respectively. Industrial activities, such as metal smelting, electroplating, and chemical and machinery manufacturing, release large amounts of heavy metals, and the dyes and auxiliaries used in textile production mostly contain heavy metals [64,65]. Given the above, Suzhou should intensify supervision and control industrial sources, especially for non-ferrous metal smelting and electroplating, to prevent any further increases in heavy metals in the soil.

### 4.3. Soil Quality Analysis and Suggestions

The results of the soil quality assessment revealed that to build high-quality farmland, Suzhou needs to focus on improving the fertility of the farmland soil. With limited arable land resources and high pressure on grain supply in Suzhou, it is hard to ensure a high grain yield without the input of fertilizers. However, the excessive application of nitrogen fertilizer caused soil acidification, and the available contents of heavy metals such as Cd and Pb also increased with the decreasing pH value. The application of phosphorus fertilizer may increase the content of bio-available Cd in soil, making it more easily absorbed by plants and leading to a higher concentration of Cd in agricultural products [66]. Furthermore, chemical fertilizers with high Cd and livestock manure with high As, Zn, and Cu bring large amounts of heavy metals into farmland soils during long-term application [67,68,69]. Therefore, to improve the quality of farmland soil, beyond strengthening the supervision and management of government departments, there is an approach to implement site-specific agricultural measures under the premise of ensuring ecological and environmental security. Measures include: (1) maintaining the existing soil protection measures to continue to promote straw returning and the action of zero growth of chemical fertilizer and pesticide use; (2) popularizing soil testing and formulated fertilization technology, where the amount, method, and period of fertilization should be optimized according to the soil nutrient status and crop growth characteristics; (3) increasing the area planted with green manure, as the application of green manure can help to maintain soil fertility and biodiversity [70]; (4) choosing fertilizers reasonably and reducing or avoiding the use of fertilizers with heavy metals; (5) composting organic fertilizer by using aquatic plants. Suzhou, with its numerous lakes, has huge amounts of aquatic plant remnants, which could be used to compost organic fertilizers [71].

## 5. Conclusions

In our study, a comprehensive assessment considering both soil fertility and the soil environment was realized in the assessment of farmland soil quality. Regarding the spatial patterns and values of the SFI, SEI, and SQI, the soil fertility of most farmlands in Suzhou was not optimistic, low-fertility soil, and the low values of the SFI were mainly concentrated in central Suzhou, Kunshan, and Taicang. Most farmland soils were not polluted by heavy metals, and the low values of the SEI were mainly concentrated in central Suzhou. With the un-even distribution of soil fertility and soil pollution with heavy metals, Cd and Pb were relatively serious, half of the farmland soil quality in Suzhou city needed to be further improved, and low values of SQI were mainly concentrated in the central city and most areas of Kunshan and Taicang. Our assessment also suggested that the soil fertility level had great influence on the results of the soil quality assessment. In the future, the government should focus on improving soil fertility to improve the quality of farmland soil and to achieve the construction of high-quality farmlands. The assessment work presented an innovative and reliable method for the comprehensive assessment of soil quality; this method has the potential to be applied in other similar areas, especially areas with a high degree of industrialization and high pressure on land resources. In addition, the method combining fertility and environmental pollution is valuable for agricultural production and soil ecological protection.

## Figures and Tables

**Figure 1 ijerph-19-12912-f001:**
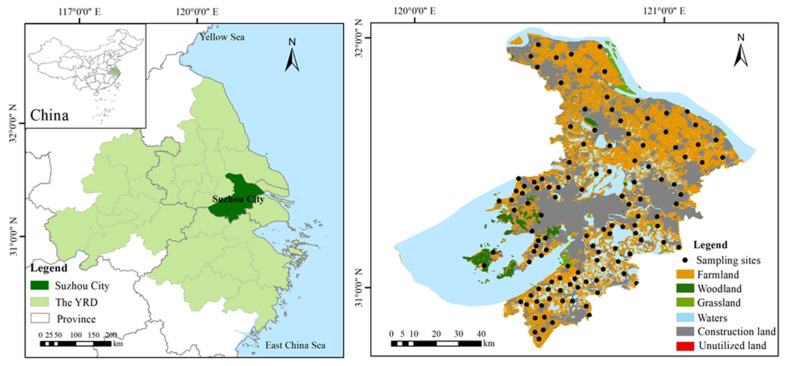
Study area and sampling sites.

**Figure 2 ijerph-19-12912-f002:**
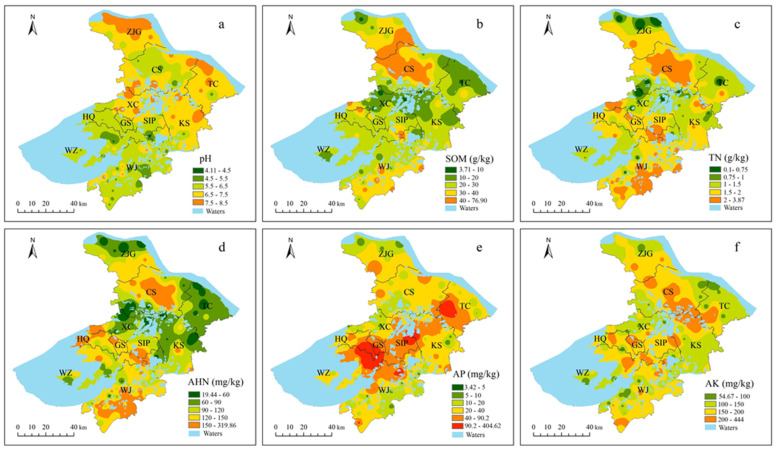
Spatial distribution of pH (**a**), SOM (**b**), TN (**c**), AHN (**d**), AP (**e**), and AK (**f**) contents in soils. The map was drawn according to soil pH and soil fertility indicator grading standards (Table 1). GS: Gusu district; HQ: Huqiu district; WZ: Wuzhong district; XC Xiangcheng district; WJ: Wujiang district; SIP: Suzhou industrial park; ZJG: Zhangjiagang; CS: Changshu; TC: Taicang; KS: Kunshan.

**Figure 3 ijerph-19-12912-f003:**
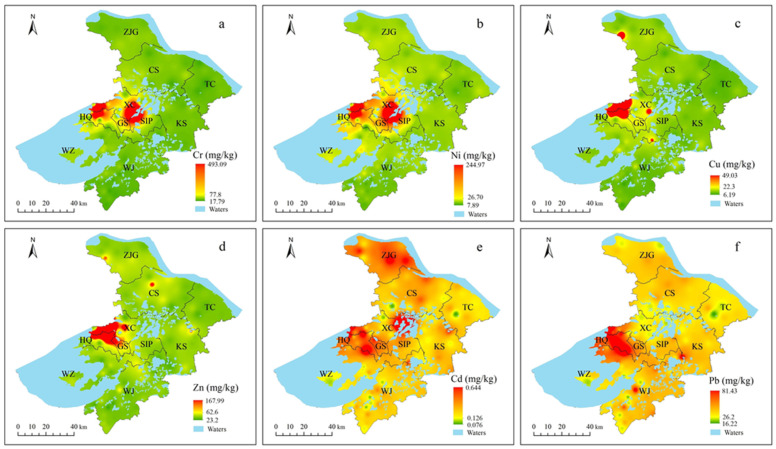
Spatial distribution of Gr (**a**), Ni (**b**), Cu (**c**), Zn (**d**), Cd (**e**), and Pb (**f**) concentrations in soils.

**Figure 4 ijerph-19-12912-f004:**
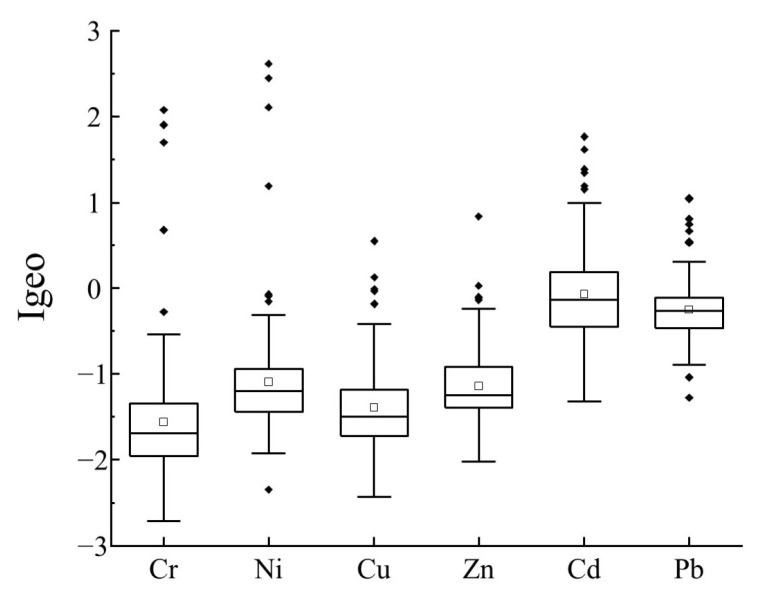
Boxplots of the geo-accumulation index ($I_{geo}$) for heavy metals in soil samples.

**Figure 5 ijerph-19-12912-f005:**
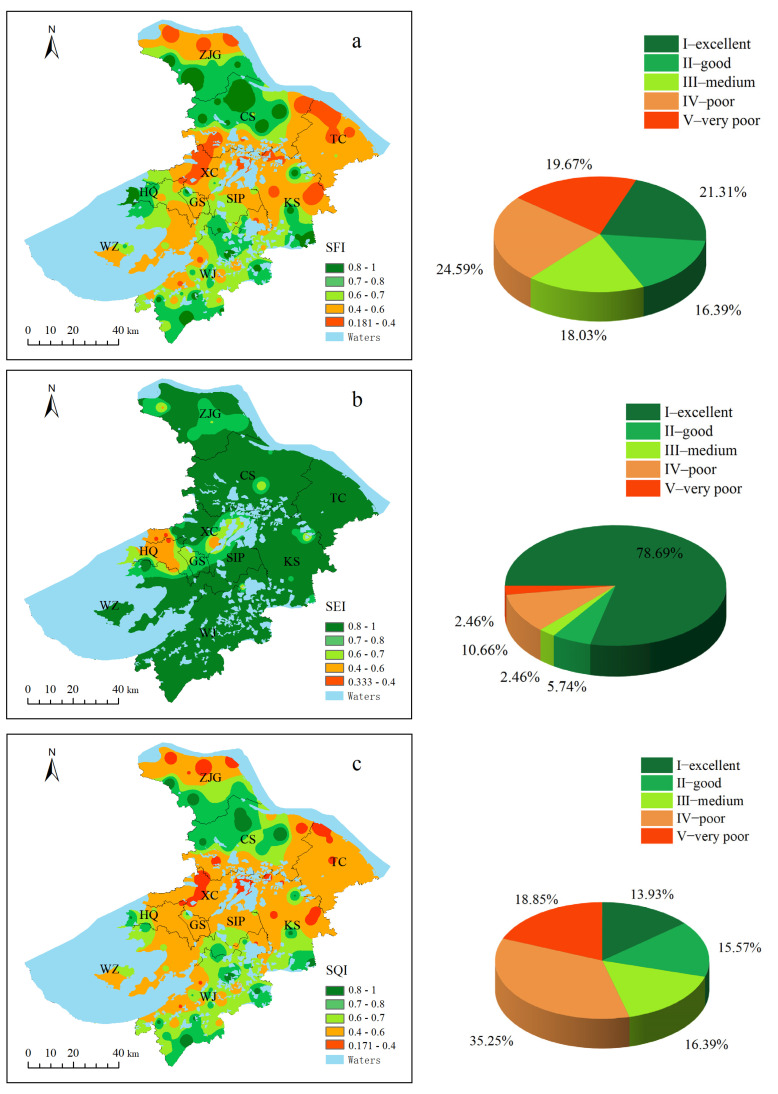
Results of the soil fertility index (SFI), soil environment index (SEI), and soil quality index (SQI). SFI (**a**), SEI (**b**), and SQI(**c**).

**Figure 6 ijerph-19-12912-f006:**
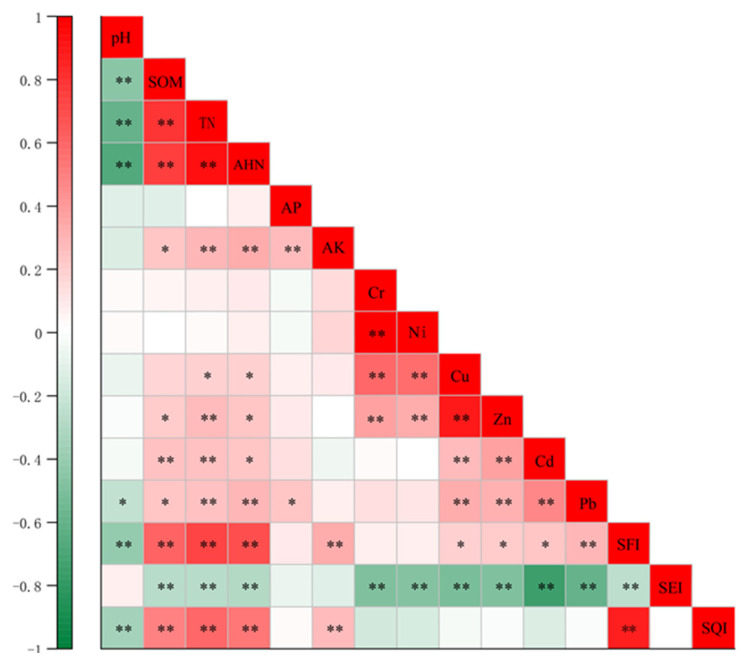
Correlation analysis between different soil indicators. * Significant at the *p* < 0.05 level. ** Significant at the *p* < 0.01 level.

**Table 1 ijerph-19-12912-t001:** Classification standards used for soil fertility indicators.

Indicator	I	II	III	IV	V
pH	6.0–7.0	6.0–5.5	5.5–5.0	5.0–4.5	≤4.5
		7.0–7.5	7.5–8.0	8.0–8.5	≥8.5
OM (g/kg)	>40.0	30.0–40.0	20.0–30.0	10.0–20.0	<10
TN (g/kg)	>2	1.5–2.0	1.0–1.5	0.75–1.0	<0.75
AHN (mg/kg)	>150	120–150	90–120	60–90	<60
AP (mg/kg)	>40.0	20.0–40.0	10.0–20.0	5.0–10.0	<5
AK (mg/kg)	>200.0	150.0–200.0	100.0–150.0	50.0–100.0	<50

**Table 2 ijerph-19-12912-t002:** Thresholds for the membership function of soil fertility indicators.

Indicator	L	U	$O_{1}$	$O_{2}$
pH	4.5	8.5	6.0	7.0
OM (g/kg)	10.0	30.0		
TN (g/kg)	0.75	1.5		
AHN (mg/kg)	60	120		
AP (mg/kg)	5.0	20.0		
AK (mg/kg)	50.0	150.0		

**Table 3 ijerph-19-12912-t003:** Descriptive statistics of soil quality indicators in the study area.

Indicators		pH	SOM	TN	AHN	AP	AK	Cr	Ni	Cu	Zn	Cd	Pb
Minimum		4.11	3.71	0.10	19.44	3.42	54.67	17.79	7.89	6.19	23.20	0.076	16.22
Twenty-fifth percentile		5.53	18.47	1.09	72.92	11.04	101.83	29.92	14.75	10.10	35.78	0.139	28.47
Fiftieth percentile		6.18	27.28	1.59	112.30	19.08	150.67	36.22	17.42	11.82	39.53	0.172	32.72
Seventy-fifth percentile		7.42	35.54	2.19	158.47	36.49	195.75	45.89	20.93	14.83	49.96	0.215	36.52
Maximum		8.34	76.90	3.87	319.86	404.62	444.00	493.09	244.97	49.03	167.99	0.644	81.43
Mean		6.44	28.17	1.63	118.16	38.31	160.63	49.44	23.43	13.71	44.92	0.196	34.35
CV (%)		16.20	51.03	44.12	49.11	165.95	48.16	130.84	134.71	45.50	39.09	48.09	31.06
Background value ^a^		/	/	/	/	/	/	77.80	26.70	22.30	62.60	0.126	26.20
Risk screening values ^b^	pH ≤ 5.5	/	/	/	/	/	/	150	60	50	200	0.3	70
	5.5 < pH ≤ 6.5	/	/	/	/	/	/	150	70	50	200	0.3	90
	6.5 < pH ≤ 7.5	/	/	/	/	/	/	200	100	100	250	0.3	120
	pH > 7.5	/	/	/	/	/	/	250	190	100	300	0.6	170
Percent ^c^ (%)		/	/	/	/	/	/	5.74	8.20	8.20	10.66	86.07	84.43
Percent ^d^ (%)		/	/	/	/	/	/	3.28	3.28	0	0	8.20	0.82

^a^ Background value for heavy metals in the topsoil of Jiangsu province, China. ^b^ The risk screening values for soil contamination of agricultural land in the soil environmental quality risk control standard for soil contamination of agricultural land (GB 15618–2018) [38]. ^c^ The percentage of soil samples exceeding the background value. ^d^ The percentage of soil samples exceeding the ‘risk screening values’.

## Data Availability

Not applicable.

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
