# Peer review of "Soil Quality Assessment in Farmland of a Rapidly Industrializing Area in the Yangtze Delta, China"

_ijerph, 2022, doi:10.3390/ijerph191912912_

Round 1
Reviewer 1 Report
***COMMENTS TO AUTHORS***
OVERALL
The study sounds local, which may lead to its rejection from scientific journals. There is nothing wrong with focusing on an area as well as promoting local cataloguing and improvement of soils. However, you need to modify/reformulate certain sections such that you demonstrate the (potential) benefit and applicability to the international audience. This will result in changes in multiple sections, which you should meticulously do. See below for further information.
Originality and importance must be stated. See below.
The maps may need to be reworked, which will change results, discussion, and some conclusions. See multiple comments below.
‘DISCUSSION’ needs several improvements. See below.
LANGUAGE QUALITY
Fair; avoid sharp, non-scientific expressions such as ‘extremely critical’ or ‘worrying’ throughout the text. Please example your revisions in your response.
TITLE
OK, but may be revised to emphasize the originality of the study.
ABSTRACT
L13 – I don’t agree. Please rephrase/delete, you indeed explained this well in Intro.
L16 – any specific reason for working this area?
L19 – not needed as you don’t say the reader whether these values are favorable or not.
INTRODUCTION
L69 – you chose this area, are similar challenges relevant to other parts of the world? Is such a study approach beneficial to the international audience?
The study originality and importance are not mentioned at all.
ORIGINALITY AND IMPORTANCE
This is missing. More clearly, it is there, but you haven’t described. Regarding importance: What are other land assessment methods and what are their weaknesses? Are there review papers in the area worth mentioning? Why did you combine the methods you picked and what is your rationale for this? What have you done in this paper that is original, tell it clearly. What is the importance and INTERNATIONAL relevance? Give these right before the objectives.
MATERIALS AND METHODS
L108 – sentence to improve.
L117 – the soil samples were digested, not the HMs.
Sort elements alphabetically. Provide rationale for selection (may be given here or in Intro).
Detection limits needed.
L123 – what were the results, then?
L125 – what are these requirements?
RESULTS
Overall, avoid/minimize location based results reporting and discussion; it is OK to a point but you have passed it. You may rather focus on similarities, correlations, scientific discussion related to mobility and toxicity, potential sources & solutions of identified soil quality problems etc. Please report all changes in your response.
3.1. The background values for the majority of elements seem low (I am not saying that they are wrong). Could you please check and verify this (the source) is proper? If needed, please switch to a more proper value set from another study. Probably they are OK. But please verify.
3.2. You must include the mapping methodology in SECTION 2. You should demonstrate that you have used a proper mapping method with appropriately selected parameters. Otherwise, the spatial distribution can’t be presented and discussed. If you used IDW, it is mostly not an appropriate method in environmental assessment and you must switch to something else (preferably Kriging).
L275: should be ‘3.2’
L288: should be ‘3.3’ and ‘in’
3.3. Again: You must include the mapping methodology in SEC 2. You should demonstrate that you have used a proper mapping method with appropriately selected parameters. Try not to use IDW.
3.4. IDW is generally not an appropriate method for environmental mapping. If it is in this case, you must demonstrate (ArcGIS can do it). Or use a different method.
If changes in mapping are significant, several places in the article will require revisions. Please provide them all in your response, if that is the case.
DISCUSSION
I invite the authors to better understand the mechanisms governing metals mobility and retention in soils. They may start with this excellent document: https://www.epa.gov/remedytech/behavior-metals-soils focusing on their selected elements. Also, here is a good example paper regarding how to discuss: https://www.sciencedirect.com/science/article/abs/pii/S2352009420300304 sections 3.2 to 3.4 may be useful to the authors in reformulating their discussion.
· L333 – it is ‘correlation’, same in L335
· L338 – there is a better another reason for this correlation. Please elaborate.
· L340 – seems plausible, any support from the literature? There may be also another reason for this. Please elaborate.
· L347 - There may be also another reason for this. Please elaborate.
L349 – this paragraph is a bit too much speculation based on the results of correlation analysis. Indeed, a sensitivity analysis for the model is needed to reach such conclusions. Please consider revising.
L398 – sort measures by expected efficiency.
CONCLUSION
OK except only of significance at regional level. After you have modified/reformulated certain sections by previous recommendations such that you demonstrate the (potential) benefit and applicability to the international audience, and then you have revised the text accordingly, please revise/rewrite the conclusion as well.
REFERENCES
Appropriate.
TABLES & FIGURES
T3 – please try to stick to three significant digits.
F4 – add abbreviation ‘(Igeo)’ to caption.
F5 – please improve caption by defining SFI SEI and SQI
F1 – ‘distribution map’ not needed
Author Response
Sept 19, 2022
Dear Editor and Reviewer:
We sincerely thank you very much for your kind help and the reviewers’ concerning our manuscript entitled “Soil Quality Assessment in Farmland of a Rapidly Industrializing Area in the Yangtze Delta, China” (Manuscript ID: ijerph-1895581). Those comments are all valuable and very helpful for revising and improving our paper, as well as the important guiding significance to our research. We have addressed the comments point by point and have made changes in the revised manuscript which we hope to meet with your approval. Revised portions are marked in red in the revised manuscript. The main corrections in the revised manuscript and the responses to the reviewer’s and editor's comments are in the attachment.

Reviewer 2 Report
This paper deals with the assessment of soil quality in agricultural areas in the Yangtze Delta. Work done well and thoroughly. The use of methods is adequate for the purpose of the work. The results obtained are reliable and well presented. A few minor comments are presented below.
1. avoid using title words as keywords
2. please provide general information on soil types and natural vegetation in this delta
Author Response
Sept 19, 2022
Dear Editor and Reviewer:
We sincerely thank you very much for your kind help and the reviewers’ concerning our manuscript entitled “Soil Quality Assessment in Farmland of a Rapidly Industrializing Area in the Yangtze Delta, China” (Manuscript ID: ijerph-1895581). Those comments are all valuable and very helpful for revising and improving our paper, as well as the important guiding significance to our research. We have studied the comments carefully and made corrections which we hope to meet with your approval. Revised portions are marked in red in the revised manuscript. The main corrections in the revised manuscript and the responses to the reviewer’s and editor's comments are in the attachment.

Round 2
Reviewer 1 Report
Thanks for the extensive revisions. The paper is greatly improved. There are two major points and one minor point that remains to be addressed:
MAJOR
1 - Section 2.3 and related sections in Section 3: You prepared a detailed rebuttal regarding the use of IDW vs. Kriging. I understand your points. That being said, I have to say that still find other interpolation methods a better predictors than IDW. In your specific case, isolated ‘islands’ of higher/lower predictions in all figures convince me that IDW is not appropriate. You are free to use whatever interpolation algorithm you would find appropriate, but then you have to do one thing, as explained below:
If you stick to this interpolation method, you have to show that it properly predicts the results. There are methods for it. Simply find it and do it. It’s quick.
- - If the validation results are good, you are done.
- - If they are not good, that means IDW is not working. In this case, play with the parameters of IDW until it works. If it does not, you need to abandon IDW and switch to something else. (Indeed, even if you switch to Kriging or something else, you have to do it. It is a part of the process. Without it, you cannot claim that your results are correct).
- - If the results change because you changed IDW parameters or you switched the method, you need to revise the manuscript accordingly.
2 - L145. You must briefly describe your methodology and then present your selected parameters for IDW (or for something else if you switch). Providing one general sentence is not enough. This is very important for results reproducibility.
MINOR
L135. The DLs for ICP seems to me wrong. I think the unit is mistaken, it may be micrograms/L, not ng/L. Please check and correct if necessary.
Author Response
Sept 29, 2022
Dear Editor and Reviewer:
We sincerely thank you very much for your kind help and the reviewers’ concerning our manuscript entitled “Soil Quality Assessment in Farmland of a Rapidly Industrializing Area in the Yangtze Delta, China” (Manuscript ID: ijerph-1895581). We appreciate the reviewer’s careful work and thoughtful suggestions and agree with the reviewer’s comments on the shortcomings of our research. Those comments are all valuable and very helpful for revising and improving our manuscript, as well as the important guiding significance to our research. We have studied the comments carefully and made corrections which we hope to meet with your approval.
Wish you all the best in the future.
The revised part is marked in red in the revised manuscript. For the main corrections in the revised manuscript and responses to the reviewers' comments, please see the attachment.

Round 3
Reviewer 1 Report
Thanks for the revisions. Although I still have some reservations about mapping, I think that the manuscript has been sufficiently improved to warrant its publication.